# Dental Pulp Inflammation Initiates the Occurrence of Mast Cells Expressing the α_1_ and β_1_ Subunits of Soluble Guanylyl Cyclase

**DOI:** 10.3390/ijms24020901

**Published:** 2023-01-04

**Authors:** Yüksel Korkmaz, Markus Plomann, Behrus Puladi, Aysegül Demirbas, Wilhelm Bloch, James Deschner

**Affiliations:** 1Department of Periodontology and Operative Dentistry, University Medical Center, Johannes Gutenberg University Mainz, 55131 Mainz, Germany; 2Center for Biochemistry, Faculty of Medicine, University of Cologne, 50931 Cologne, Germany; 3Department of Oral and Maxillofacial Surgery, University Hospital RWTH Aachen, RWTH Aachen University, 52074 Aachen, Germany; 4Department of Restorative Dentistry, Faculty of Dentistry, Mugla Sıtkı Kocman University, Mugla 48000, Turkey; 5Department of Restorative Dentistry, Faculty of Dentistry, Ege University, Izmir 35040, Turkey; 6Department of Molecular and Cellular Sport Medicine, German Sport University Cologne, 50933 Cologne, Germany

**Keywords:** inflammation, dental pulp, pulpitis, mast cells, nitric oxide, soluble guanylyl cyclase, cGMP, ROS, peroxynitrite, RNS

## Abstract

The binding of nitric oxide (NO) to heme in the β_1_ subunit of soluble guanylyl cyclase (sGC) activates both the heterodimeric α_1_β_1_ and α_2_β_1_ isoforms of the enzyme, leading to the increased production of cGMP from GTP. In cultured human mast cells, exogenous NO is able to inhibit mast cell degranulation via NO-cGMP signaling. However, under inflammatory oxidative or nitrosative stress, sGC becomes insensitive to NO. The occurrence of mast cells in healthy and inflamed human tissues and the in vivo expression of the α_1_ and β_1_ subunits of sGC in human mast cells during inflammation remain largely unresolved and were investigated here. Using peroxidase and double immunohistochemical incubations, no mast cells were found in healthy dental pulp, whereas the inflammation of dental pulp initiated the occurrence of several mast cells expressing the α_1_ and β_1_ subunits of sGC. Since inflammation-induced oxidative and nitrosative stress oxidizes Fe^2+^ to Fe^3+^ in the β_1_ subunit of sGC, leading to the desensitization of sGC to NO, we hypothesize that the NO- and heme-independent pharmacological activation of sGC in mast cells may be considered as a regulatory strategy for mast cell functions in inflamed human dental pulp.

## 1. Introduction 

The inter- and intracellular molecule nitric oxide (NO)-sensitive soluble guanylyl cyclase (sGC) is a heterodimeric enzyme with an α and β subunit [1,2]. The redox state of the heme moiety of the β_1_ subunit of sGC is critical for the NO–heme iron interaction within the β_1_ subunit in order to activate sGC heterodimers [3,4]. Heme iron in the reduced state (Fe^2+^) is required for the binding of NO to the β_1_ subunit of sGC, whereas heme iron in the oxidized state (Fe^3+^) causes the enzyme to be insensitive to NO and decreases cGMP production [3,4,5,6]. Under physiological conditions, NO binds to heme in the β_1_ subunit of sGC and activates the enzyme containing reduced Fe^2+^ in the α_1_β_1_ and α_2_β_1_ isoforms. In the heterodimeric α_1_β_1_ and α_2_β_1_ isoforms, sGC induces the production of cGMP, resulting in cell-specific functions such as vasodilation [3,7], the inhibition of vascular smooth muscle proliferation [8,9], leukocyte recruitment and platelet aggregation [9,10], and the modulation of neurotransmission [11,12].

Mast cells originate from the yolk sac during embryogenesis and after birth from CD34+ hematopoietic stem cells in the bone marrow via definitive hematopoiesis [13,14]. Immature mast cells enter the tissues and undergo their final maturation into connective tissue mast cells or mucosal mast cells after receiving an appropriate signal from the tissue environment [14,15]. Mast cells are activated by bacteria and their products via innate immunity through pattern recognition receptors, such as TLRs and NOD-like receptors, expressed in mast cells [16,17] and promote adaptive immunity by modulating dendritic cell migration and immunological cellular processes in lymph nodes [17,18,19]. In adaptive immunity, mast cells are sensitized by different classes of immunoglobulins via Fc receptors (FcRs) [19,20]. Mast cells are activated when IgE bound to FcεRI receptors is crosslinked by bi- or multivalent antigens [21,22]. Upon activation by their receptors, mast cells release histamine, tryptase, and chymase from their secretory granules via degranulation [21,23]. Mast cells produce lipid mediators (prostaglandins and leukotrienes) as well as proinflammatory and chemotactic cytokines (tumor necrosis factor (TNF), interleukin (IL)-6, IL-4, IL-5, IL-1β, IL-10, IL-13, CCL1, CCL2, CXCL1, and CXCL8) [21,23]. The mediators secreted by mast cells contribute to host immunity and inflammation associated with various diseases [20,23,24].

NO is produced in mast cells through the activity of eNOS, nNOS, and iNOS [25,26,27]. In cultured mast cells, NO is able to inhibit mast cell degranulation [28,29] via the NO–cGMP signaling cascade [30]. NO donors inhibit histamine release and mast cell degranulation, whereas NO inhibitors enhance LPS-induced histamine release in mast cells [28]. The role of sGC in mast cells was investigated in cultured mast cells using the inhibitor 1H-[1,2,4]oxadiazolo [4,3-a]quinoxalin-1-one (ODQ), specific for sGC [30]. In cultured human mast cells, it was found that NO donors were able to increase the formation of cGMP through the release of exogenous NO [30]. However, the inflammation-dependent formation of superoxide (O_2_^−^) and peroxynitrite (ONOO^−^) oxidizes sGC, leading to the insensitivity of sGC to NO [3,5]. Therefore, under inflammatory conditions, sGC may only be activated in an NO- and heme-independent manner. Since the pharmacological NO- and heme-independent activation or stimulation of sGC requires the presence of the enzyme at the protein level [31,32,33,34], it is necessary to clarify whether sGC occurs at the protein level in mast cells in tissues under physiological and inflammatory conditions.

Inflammation in the dental pulp occurs as a consequence of bacterial infection caused by dentin caries [35,36,37,38]. The carious bacteria and their products reach the dental pulp through the dentinal tubules, initiating an immunological host defense reaction in the dental pulp [39,40]. Host defenses in the dental pulp are regulated by the innate and adaptive immune responses [40,41,42,43], and mast cells are involved in their regulation upon bacterial infection [17,18,24,44,45,46]. The presence of mast cells in healthy [47] and inflamed [48] dental pulp has been described previously. However, the occurrence and the role of mast cells in healthy dental pulp and in pulp necrosis as well as chronic and acute pulp inflammation remain to be elucidated. Therefore, in the present study, we addressed the presence of mast cells and the α_1_ and β_1_ subunits of sGC in these cells in healthy and inflamed dental pulp.

## 2. Results

In cells from healthy dental pulp (Figure 1A–D), MCT was not found (Figure 1E–H). In cells from inflamed dental pulp (Figure 2A–D), MCT (Figure 2E–H) was detected in several mast cells. In the inflamed dental pulp, the α_1_ (Figure 3A–P) and β_1_ (Figure 4A–P) subunits of sGC were colocalized with MCT in mast cells at different staining intensities (Figure 5).

### 2.1. Characterization of the Healthy Human Dentin–Pulp Complex and the Expression of Mast Cell Tryptase (MCT) in Cells from Healthy Human Dental Pulp

In the healthy dentin–pulp complex, primary dentin, secondary dentin, and predentin were seen in a structural order (Figure 1A,B and Appendix A). The odontoblast layer, the cell-free and cell-rich layers, and the subodontoblastic plexus with nerve fibers and blood vessels were also identified in a cellular order in the healthy dentin–pulp complex (Figure 1A,B and Appendix A). The healthy dental pulp contained numerous cells (Figure 1C,D and Appendix A). Several pulpal stromal cells were identified around the blood vessels and nerve fibers (Figure 1C,D and Appendix A). Arterial and venous blood vessels of different sizes (arteries, veins, and capillaries) were accompanied by nerve fibers in the dental pulp (Figure 1B–D and Appendix A).

In cells of the healthy dental pulp, the immunohistochemical localization of MCT was not identified (Figure 1E–H).

### 2.2. Characterization of Inflamed Human Dentin–Pulp Complex and the Expression of Mast Cell Tryptase (MCT) in Cells from Inflamed Human Dental Pulp

The structural order found in the healthy dentin–pulp complex (Figure 1A–D and Appendix A) was not visible in the dentin–pulp complex inflamed by a carious lesion (Figure 2A–D and Appendix A). The deep dentin caries destroyed the primary and secondary dentin layers (Figure 2A and Appendix A). Beneath the carious lesion, tertiary dentin was found (Figure 2A,B and Appendix A). A strong lymphocytic infiltrate with numerous inflammatory cells was detected in the dental pulp underlying the carious region (Figure 2A,B and Appendix A). In the acute and subacute inflammatory areas, numerous neutrophilic granulocytes emerged from the dilated blood vessels (Figure 2C and Appendix A), whereas the blood vessels in the more lymphocytic inflammatory areas were destroyed and degraded (Figure 2D and Appendix A). In several cases, inflammation in the dental pulp occurred in mixed forms, e.g., chronic, subacute, and acute inflammation (Figure 2C,D and Appendix A).

Beneath the carious lesion in the dental pulp, some lymphocytic inflammatory areas were identified (Figure 2A,B and Appendix A). Necrotic changes, including destroyed blood vessels, were found in the center of some inflammatory areas (Appendix A). Around these necrotically altered inflammation areas, severe acute and chronic inflammatory areas with numerous different inflammatory cells and destroyed blood vessels were observed (Appendix A). In these acute and chronic inflammatory regions, mast cell tryptase (MCT) was detected in several mast cells showing a larger cytoplasmic area and larger nuclei with avidin-biotin-peroxidase staining (Figure 2E–H and Appendix A). The localization of MCT in mast cells was confirmed in chronically inflamed dental pulp by immunofluorescence staining (Appendix A).

### 2.3. Expression of the α_1_ Subunit of sGC in Mast Cells from Inflamed Dental Pulp

The numerous inflammatory cells detected by nuclear staining with DRAQ5 showed severe inflammation in the dental pulp (Figure 3A,E,I,M). MCT was identified in a small subpopulation of inflammatory cells showing a larger cytoplasm and nucleus than the other inflammatory cells (Figure 3B,F,J,N). In some mast cells, the sGC α_1_ subunit was detected with strong staining intensities (Figure 3C,G,K), while in other mast cells, the staining intensity was weak (Figure 3C and Figure 4O). Upon colocalization, the sGC α_1_ subunit was identified with subcellular MCT at both stronger (Figure 3I–L) and weaker (Figure 3M–P) staining intensities in the cytoplasm. In a very small subpopulation of immune cells, the sGC α_1_ subunit was found in the cytoplasm without colocalization with MCT (Figure 3D,H,L).

### 2.4. Expression of the β_1_ Subunit of sGC in Mast Cells from Inflamed Dental Pulp

The presence of numerous nuclei of inflammatory cells detectable by DRAQ5 indicated severe inflammation in the dental pulp (Figure 4A,E,I,M). MCT was detected in a small subpopulation of inflammatory cells showing a large cytoplasm and nuclei, whereas the other inflammatory cells contained small and round nuclei (Figure 4B,F,J,N). In some inflammatory cells, the sGC β_1_ subunit was identified with MCT at a weak staining intensity (Figure 4C), while a stronger intensity was detected in some other inflammatory cells (Figure 4C,G,K,O). In the double stainings, the sGC β_1_ subunit in mast cells colocalized with MCT in one subpopulation (Figure 4D,H,L,P), whereas no colocalization was observed in the other subpopulations of inflammatory cells (Figure 4D,H,L).

### 2.5. Immunohistochemical Controls

In control incubations of the avidin–biotin–peroxidase complex (Appendix A) and the immunofluorescence double staining methods (Appendix A), no immunohistochemical staining was found in cells from healthy and inflamed human dental pulps.

### 2.6. Quantification of the Data and Statistical Analysis

#### 2.6.1. Counting of Mast Cells in Healthy and Inflamed Dental Pulp

MCT was not detectable in any cells of the healthy human dental pulp (*n* = 6). In the inflamed human dental pulp (*n* = 6), MCT was detected in mast cells. When compared, the inflamed dental pulp showed different numbers of mast cells (cell counts 111 ± 16 [mean ± SEM]; 51–296 [range]), which can be explained by the different mixed inflammation states in the dental pulp.

#### 2.6.2. Staining Intensities of the α_1_ and β_1_ Subunits of sGC in Mast Cells

The immunofluorescence staining intensities for the α_1_ and β_1_ subunits of sGC showed similar results (Figure 5). No significant differences were found between the staining intensities of the α_1_ and β_1_ subunits of sGC in mast cells from the inflamed human dental pulp (*p* = 0.22).

## 3. Discussion

Histopathological studies of mast cells in inflamed tissues are required to elucidate their pathophysiological role in organ-specific inflammation [24]. Although NO production has been shown in mast cells [25,27], it is unclear whether NO-sensitive sGC occurs as an active heterodimer isoform in these cells under physiological and inflammatory conditions. In our study, we found that mast cells were not present in healthy dental pulps and only appeared in inflamed dental pulps. In severely inflamed dental pulps, we detected the expression of both the α_1_ and β subunits of sGC in mast cells, indicating the existence of the α_1_β_1_ heterodimers of sGC at the protein level in mast cells during inflammation. In inflammation, O_2_^−^ and ONOO^−^ oxidize sGC (Fe^2+^ to Fe^3+^ in the β_1_ subunit of sGC), leading to the insensitivity of sGC to NO [3,4,5,6]. Because O_2_^−^ and ONOO^−^ have been detected at higher concentrations in inflamed dental pulp [49,50], we suggested that sGC might be present as an α_1_β_1_ isoform in mast cells at the protein level and could be insensitive to NO in the oxidized state in these cells during the inflammation of human dental pulp.

In the present study, the inflammatory regions in human dental pulp showed the histopathological features of mixed inflammation, ranging from a necrotic region to regions with chronic, subacute, and acute inflammatory features. In necrotically altered inflammatory regions, where blood vessels were degraded, no mast cells were found. In inflamed dental pulps, mast cells were observed around blood vessels in chronic, subacute, and/or acute inflammatory regions. In chronically inflamed regions, numerous neutrophil granulocytes left the blood vessel walls and formed acute inflammatory areas in the pulp (Figure 2C and Appendix A). Due to their role as the first effector cells recruited to sites of inflammation, neutrophils are crucial cells in the regulation of acute immune responses during inflammation [51,52,53,54]. In inflammation, mast cells orchestrate immunity against pathogens by secreting cytokines that trigger neutrophil granulocyte recruitment [19,55,56,57,58]. Our results showed that mast cells in inflamed dental pulps strongly expressed the protease mast cell tryptase. Mast cell tryptase has been shown to be involved in neutrophil granulocyte recruitment during inflammation [59,60]. Therefore, it is reasonable to assume that neutrophil granulocyte recruitment during dental pulp inflammation may be regulated in part by mast cell tryptase.

The mechanism controlling the inflammatory response [61] in dental pulp is not well understood. Inflammation triggers the complex processes of innate and adaptive immunity that serve to resolve inflammation in the acute phase [62,63]. Acute inflammation occurs over a few days or weeks and requires the presence of an external stimulus [63]. However, with prolonged or more intense infiltration by various immune cells, acute inflammation can progress to chronic inflammation [63,64]. Chronic inflammation may persist well beyond the presence of the external stimuli for months or years, mediated by various immune cells [63]. The impact of mast cell-derived inflammatory mediators on blood vessels may be a key driving force in the initiation, enhancement, or maintenance of acute and chronic inflammation [65]. In acute bacterial infections, the activity of mast cells leads to the early clearance of bacteria and resolution of inflammation [45,66]. In chronic inflammation, the modulation of inflammation by mast cells is much more complex, as interactions between mast cells and bacteria are prolonged [19,45]. Depending on the type of bacteria and the severity of infection, mast cells may promote rather than control chronic infections and can therefore exacerbate pathologic outcomes [45,66]. Mast cells secrete inflammatory mediators, accumulate, and proliferate in the inflammatory region, where they are long-lived [19,45]. In view of these findings and our results, we hypothesize that the treatment of a carious lesion during the acute inflammatory state of the dental pulp could improve its healing tendency through the functions of mast cells. If a carious lesion is not treated in time, the acute inflammation in the dental pulp may become chronic. In chronic pulp inflammation, mast cells promote inflammation by recruiting neutrophilic granulocytes, which may lead to the development of mixed (necrosis, chronic, subacute, and acute inflammatory regions) and thus uncontrolled inflammation.

We detected the α_1_ and β_1_ subunits of sGC in the mast cells of inflamed dental pulp with different staining intensities. In some mast cells, the α_1_ and β_1_ subunits of sGC were detected with a stronger staining intensity, whereas in other mast cells the intensity was lower. The difference in the intensities of the α_1_ and β_1_ subunits of sGC suggests that mast cells exist in different functional states under inflammatory conditions and that the expression of the α_1_ and β_1_ subunits of sGC depends on the chronic, subacute, or acute inflammation region-associated mast cell behavioral status.

In smooth muscle cells and in neurons, sGC is active as both α_1_β_1_ and α_2_β_1_ heterodimers. It is possible that sGC may also be activated as α_2_β_1_ heterodimers in subpopulations of mast cells negative for α_1_ subunit of sGC. Therefore, future studies should clarify whether the α_2_ subunit of sGC is present in mast cells at the protein level under in vivo conditions. We detected the α_1_ subunit of sGC only in mast cells and in blood vessels, whereas the β_1_ subunit of sGC was detected in mast cells, blood vessels, and also in a subpopulation of other inflammatory cells that were negative for MCT. Therefore, it should be clarified whether the α_2_ subunit of sGC is also present in other immune cells that are negative for the α_1_ subunit and MCT but positive for the β_1_ subunit.

In inflammation, the consumption of molecular oxygen (O_2_) by NADPH oxidase is increased in macrophages, neutrophil granulocytes, and dendritic cells that accumulate in the area of inflammation because NADPH oxidase transfers electrons from intracellular NADPH to O_2_ to generate superoxide (O_2_^−^) [67]. The O_2_^−^ generated by uncoupled endothelial nitric oxide synthase (eNOS) [68,69] or in macrophages, neutrophil granulocytes, and dendritic cells leads to the generation of the free radical hydrogen peroxide (H_2_O_2_), which can lead to the formation of reactive oxygen species (ROS) and reactive nitrogen species (RNS) under inflammatory conditions [67,70]. The reduced Fe^2+^ state in the β_1_ subunit of sGC under physiological conditions and the oxidized Fe^3+^ state in the β_1_ subunit of sGC under inflammatory conditions are essential for determining the activity of sGC in a cell [3,4,5,6]. NO is produced in mast cells [25,27] and is able to inhibit mast cell degranulation and subsequent allergic inflammation [28,29]. NO donors inhibit histamine release and mast cell degranulation, whereas NO inhibitors enhance LPS-induced histamine release [28]. In cultured human mast cells, it has been shown that NO donors, such as DEA/NO, are able to increase the formation of cGMP through the release of exogenous NO [30], suggesting that cGMP may be formed via the activation of sGC by exogenous NO. However, during inflammation, sGC is oxidized (from Fe^2+^ to Fe^3+^) by higher concentrations of O_2_^−^ and ONOO^−^, leading to the insensitivity of sGC to NO [3,4,5,6]. Inflammation induces an increase in ROS and RNS concentrations [71,72]; this effect has been reported for inflamed dental pulp [49,50]. In addition, the inflammation-dependent synthesis of ROS and RNS in mast cells has also been described [70]. Based on these results, we suggest that higher ROS and RNS concentrations in mast cells under inflammatory conditions may oxidize the heme iron of sGC and desensitize it to NO. Thus, NO-dependent cGMP formation (cGMP may be formed also in part via particulate GC by natriuretic peptide) in mast cells under inflammatory conditions may be regulated by NO- and heme-independent sGC activators. Therefore, the use of sGC activators [31,33,34,73] and the modulation of sGC in mast cells may serve as a new treatment strategy for inflamed dental pulp.

Our results showed that mast cells were present in inflamed but not in healthy human dental pulp. In these mast cells, sGC was detectable at the protein level with both its α_1_ and β_1_ subunits. Therefore, the NO- and heme-independent pharmacological activation of sGC in mast cells may be considered as a regulatory strategy for mast cell functions in inflamed human dental pulp. For example, indirect and direct pulp capping materials used in the treatment of deep dentin caries containing sGC activators could represent an immunomodulatory treatment option in the future.

## 4. Materials and Methods

### 4.1. Ethics Statement on the Collection of Human Molars

The Human Ethics Committee of the Heinrich-Heine University Düsseldorf approved the collection of human third molars extracted during orthodontic treatment (No. 2980).

### 4.2. The Clinical Evaluation of Human Molars

The healthy molars were unrestored and clinically asymptomatic and showed no pain upon stimuli-induced testing and percussion. The carious molars showed clinically stimuli-induced, spontaneous, or percussion-induced pain and/or radiographic carious dentin and inflamed periodontal lesions.

### 4.3. Tissue Preparation

The healthy (*n* = 6) and inflamed third molars with deep dentin caries (*n* = 6) were extracted from patients who had undergone orthodontic extraction treatment. The molars were immersion-fixed in a fixative containing 4% paraformaldehyde and 0.2% picric acid in 0.1 M phosphate-buffered saline (PBS), pH 7.4, for 48 h. The molars were demineralized in 4 M formic acid for 21 days. The non-carious and carious molars were cryoprotected with 30% sucrose solution in 0.1 M PBS, pH 7.4, for 48 h. The molars were frozen-embedded, stored at −82 °C, and frozen-sectioned on a cryostat at 30 µm.

### 4.4. Histopathological Evaluation of Healthy and Inflamed Human Dental Pulp

Since the definitive diagnosis of pulpal inflammation is determined by histopathology, sections were stained with hematoxylin and eosin (HE) [49] and the results were characterized by histopathological diagnosis. In each subgroup of molars, immunohistochemical staining for mast cell tryptase (MCT), HLA-DR (monocytes, dendritic cells, and activated T and B cell markers), and CD68 (macrophage marker) were performed to characterize the leukocyte types in the healthy and inflamed human dentin–pulp complex and periodontium, as described previously [50,74].

### 4.5. Specificity of sGC α_1_ Subunit and β_1_ Subunit Antibodies

To detect the α_1_ subunit of sGC in mast cells, we developed a specific polyclonal rabbit antibody against the human α_1_ subunit of sGC (EP101278: ID0490; Eurogentec, Seraing, Belgium) that was characterized by immunohistochemistry and immunoblotting [75]. The specificity of the rabbit anti-human sGC β_1_ subunit antibody was verified with immunohistochemistry and immunoblotting using lung protein extracts from sGCβ_1_^+/+^ and sGCβ_1_^−/−^ mice [76].

### 4.6. Immunohistochemical Methods

#### 4.6.1. Avidin–Biotin–Peroxidase Complex Method

Free-floating sections were incubated with 0.3% H_2_O_2_ in 0.05 M Tris-buffered saline (TBS) for 20 min to inhibit endogenous peroxidase. Nonspecific immunoglobulin binding sites were blocked by the incubation of sections in blocking solution containing 5% normal goat serum (Vector, Burlingame, CA, USA) and 2% bovine serum albumin (BSA) (Sigma-Aldrich, Saint Louis, MO, USA). The sections of subgroups of healthy and carious molars were incubated with mouse monoclonal anti-human MCT (1:2000) (Santa Cruz Biotechnology, Santa Cruz, CA, USA), mouse monoclonal anti-human HLA-DR (1:2000) (eBioscience, San Diego, CA, USA), and mouse monoclonal anti-human CD68 (1:2000) (eBioscience). To detect the α_1_ and β_1_ subunits in cells from the healthy and inflamed dental pulps, sections were incubated overnight with rabbit anti-human polyclonal sGC α_1_ (1:1000) and β_1_ subunit (1:1000) antibodies at 4 °C. Then, sections were incubated for 1 h with biotinylated goat anti-mouse IgG (1:1000) (Vector) or goat anti-rabbit IgG (1:500) (Vector). The sections were incubated for 1 h with avidin–biotin–peroxidase complex (1:100) (Vector). The immunohistochemical reaction was developed in all incubations for 15 min with 0.05% 3,3′-diaminobenzidine tetrahydrochloride (Sigma-Aldrich, St. Louis, MO, USA) in 0.05 M Tris-HCl buffer, pH 7.6, containing 0.01% H_2_O_2_ and 0.01% nickel sulfate [49,50].

To control for secondary antibodies and avidin–biotin–peroxidase complex reagents (NGS, BSA, peroxidase complex), the omission of primary antibodies was performed as control staining.

#### 4.6.2. Immunofluorescence Double Staining Method

Free-floating sections were incubated with 5% normal goat serum and 2% BSA to block the nonspecific immunoglobulin binding sites of the secondary antibodies. The sections were incubated first with mouse anti-human MCT antibodies (Santa Cruz Biotechnology) at 4 °C. Then, the sections were incubated with DyLight^TM^ 488-conjugated goat anti-mouse IgG (Thermo Fischer Scientific, Waltham, MA, USA) for 1 h. In separate incubations, the sections were incubated with rabbit polyclonal anti-human α_1_ subunit (1:1000) and rabbit polyclonal β_1_ subunit (1:1000) antibodies overnight at 4 °C. Then, in separate incubations, the sections were incubated for 1 h with DyLight^TM^ 550-conjugated goat anti-rabbit IgG (Thermo Fischer Scientific). To show the cell nuclei, sections were stained with DRAQ5 (Cell Signaling Technology, Frankfurt am Main, Germany) for 15 min. The sections were coverslipped with Aqua Poly/Mount (Polysciences Inc., Warrington, PA, USA) and analyzed with an LSM510 confocal microscope (Carl Zeiss, Jena, Germany) [49,74].

To control for the immunohistochemical reagents (NGS, BSA, secondary antibodies), sections were incubated in the absence of the first and secondary primary antibodies.

### 4.7. Quantification of the Data and Statistical Analysis

#### 4.7.1. Counting of Mast Cells in the Inflamed Dental Pulp

In healthy (*n* = 6 molars of 6 different patients) and inflamed (*n* = 6 molars of 6 different patients) dental pulp, MCT-positive cells were visualized using conventional light microscopy. Sections stained with the avidin-biotin-peroxide method were imaged at 5× magnification using an Olympus microscope (Olympus Deutschland GmbH, Hamburg, Germany) connected to the Cell F Imaging software (Olympus Soft Imaging Solutions GmbH, Münster, Germany). Then, an examiner (B.P.) counted the MCT-positive cells using the counting tool in QuPath (version 0.3.2) [77]. The source code for QuPath is available at https://qupath.github.io. The results were verified by a second examiner (Y.K.).

#### 4.7.2. Measurement of Immunofluorescence Staining Intensities

Three colour fluorescence images were taken with an LSM 510 META confocal microscope (Carl Zeiss, Oberkochen, Germany) and exported to QuPath (version 0.3.2) [77]. The cellular localizations of the α_1_ subunit and β_1_ subunit of sGC in mast cells were annotated by one investigator (B.P.) and reviewed by a second investigator (Y.K.). The mean intensity of the red fluorescence (i.e., the staining intensities of the α_1_ subunit and β_1_ subunit of sGC) was then exported to R.

#### 4.7.3. Statistical Analysis

Statistical analyses were performed in R (version 4.1.1). Data were tested for normal distribution using a Shapiro–Wilk test. Parametric values were tested for significant *p* values using an unpaired *t*-test. A *p* value < 0.05 was considered significant. For cell counting, the mean and SEM (standard error of the mean) were calculated.

## Figures and Tables

**Figure 1 ijms-24-00901-f001:**
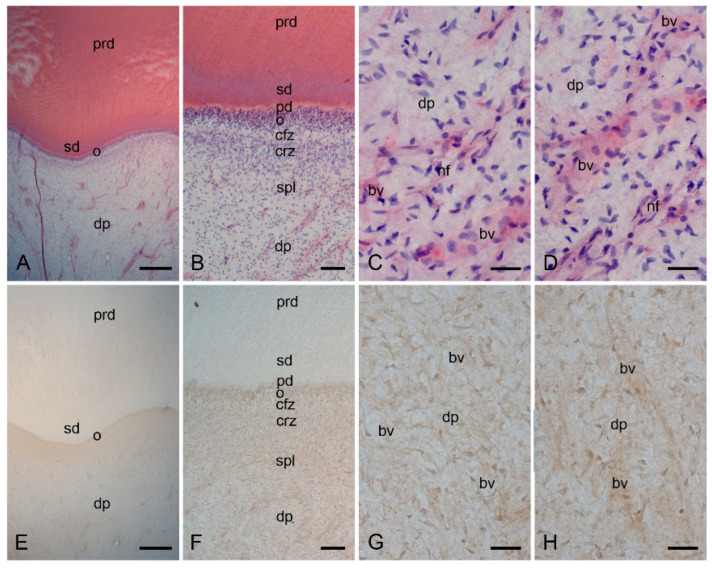
Histopathological characterization of healthy human dentin–pulp complex via hematoxylin and eosin (HE) staining and the expression of mast cell tryptase (MCT) in cells from consecutive sections of healthy human dental pulp. The structural order of the healthy dentin–pulp complex in the overview image (**A**) is visible in the detailed images as primary dentin (prd), secondary dentin (srd), predentin (pd), odontoblast layer (o), cell-free zone (cfz), and cell-rich zone (crz) (**B**), and blood vessels (bv), nerve fibers, and numerous pulp cells in the human dental pulp (dp) (**C,D**). In the cells of the healthy dental pulp of the consecutive section, the immunohistochemical localization for MCT is not visible in the overview (**E**) and detailed images (**F**–**H**). Scale bars: (**A**,**E**) = 500 µm; (**B**,**F**) = 100 µm; (**C**,**D**,**G**,**H**) = 30 µm.

**Figure 2 ijms-24-00901-f002:**
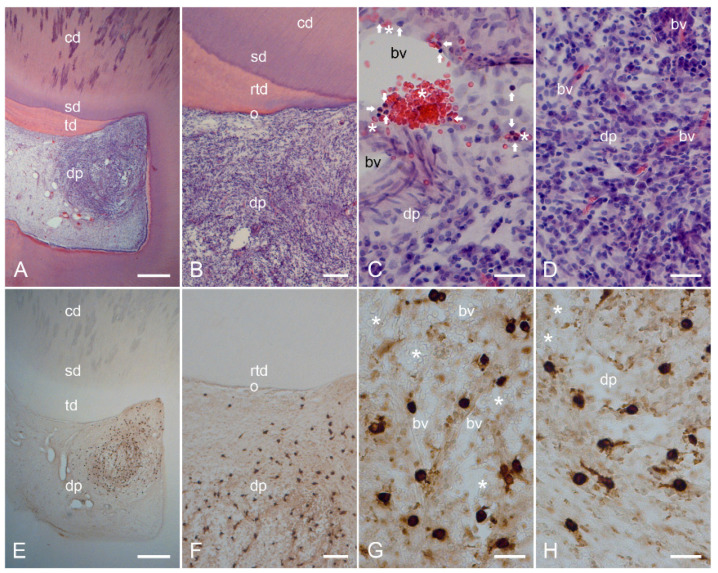
Histopathological characterization of inflamed human dentin–pulp complex by hematoxylin and eosin (HE) staining and the expression of mast cell tryptase (MCT) in cells from consecutive sections of inflamed human dental pulp. In deep dentin caries of the dentin–pulp complex, the carious dentin lesion sites (cd) with destroyed primary dentin, secondary dentin (srd), and the formation of reactive tertiary dentin (rtd) are visible (**A**). The structural and cellular order of the healthy dentin–pulp complex is not visible in the carious dentin–pulp complex (**A**,**B**). Beneath the carious lesion, there are severe chronic and acute inflammatory areas with numerous different inflammatory cells and dilated blood vessels (**B**–**D**). Neutrophilic granulocytes (arrows) associated with numerous erythrocytes (asterisks) are found in the lumen and wall of the blood vessels (**C**). In the consecutive section of the inflamed human dental pulp, MCT is detected in mast cells within chronically and acutely inflamed areas, which are seen in the overview (**E**) and detailed images (**F**–**H**). The mast cells are associated with blood vessels, which contain numerous erythrocytes (asterisks; **G**,**H**). Scale bars: (**A**,**E**) = 500 µm; (**B**,**F**) = 100 µm; (**C**,**D**,**G**,**H**) = 30 µm.

**Figure 3 ijms-24-00901-f003:**
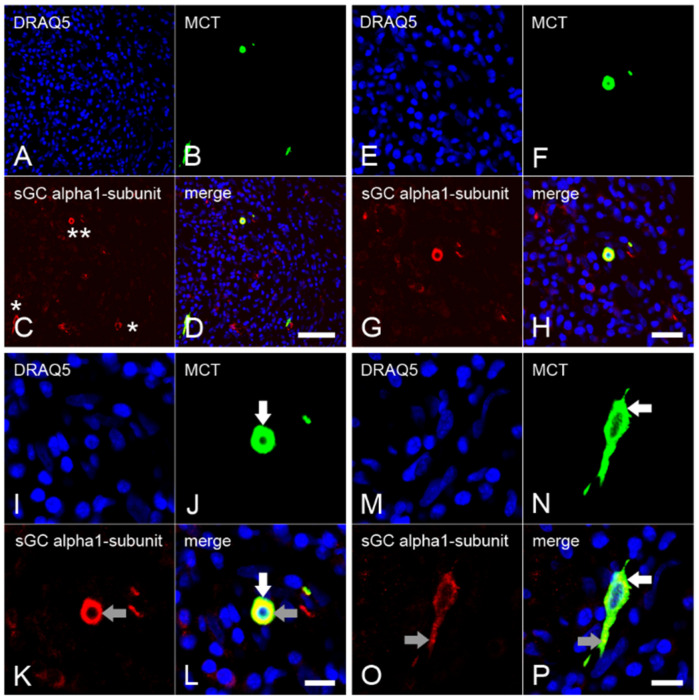
Colocalization of the α_1_ subunit of sGC with MCT in mast cells in chronically inflamed human dental pulp. Numerous inflammatory cells with DRAQ5 nuclear staining are visible in the overview images of chronically inflamed human dental pulp (**A**). MCT (**B**) is colocalized with the sGC α_1_ subunit (**C**) in some inflammatory cells with strong ((**C**), two asterisks) and weak ((**C**), one asterisk) staining intensities (**D**). Detailed images show the cytoplasmic subcellular colocalization of MCT with the sGC α_1_ subunit at strong (**E**–**L**) and weak (**M**–**P**) immunostaining intensities in mast cells. In mast cells, the localizations of MCT (**J**,**L**,**N**,**P**) are shown with white arrows, while grey arrows show the α_1_ subunit of sGC (**K**,**L**,**O**,**P**). Scale bars: **A**–**D** = 50 µm; **E**–**H** = 20 µm; **I**–**P** = 10 µm.

**Figure 4 ijms-24-00901-f004:**
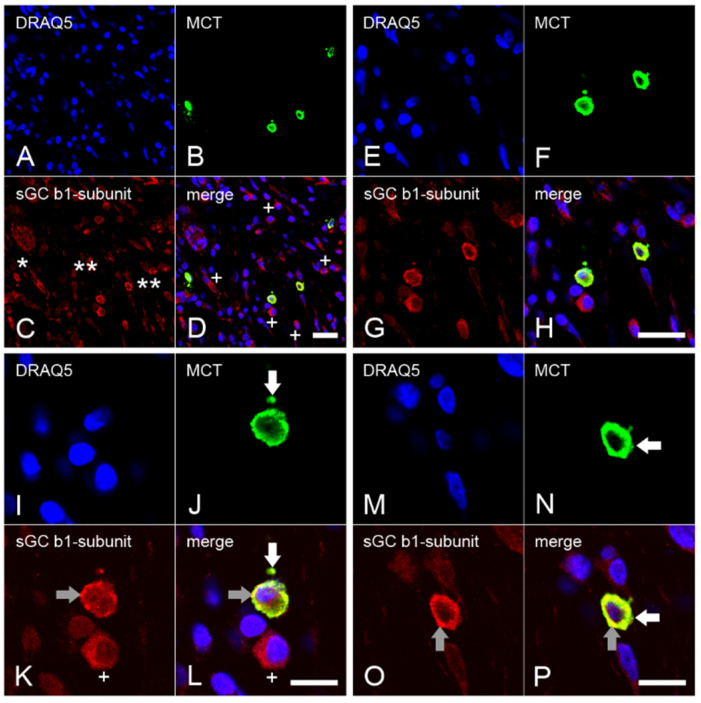
Colocalization of the β_1_ subunit of sGC with MCT in mast cells of chronically inflamed human dental pulp. Numerous inflammatory cells with DRAQ5 nuclear staining are present in the overview images of the chronically inflamed dental pulp (**A**). MCT (**B**) colocalizes with sGC β_1_ subunit (**C**) in a subpopulation of the inflammatory cells with strong ((**C**), two asterisks) and weak ((**C**), one asterisk) immunostaining intensities (**D**). In another subpopulation of cells that are MCT negative, the sGC β_1_ subunit is apparent (+ character in **D**,**K**,**L**). In the detailed images, the colocalization of MCT with the sGC β_1_ subunit in the cytoplasm of the mast cells is apparent (**E**–**P**). In mast cells, the localizations of MCT (**J**,**L**,**N**,**P**) are shown with white arrows, while the grey arrows show the β_1_ subunit of sGC (**K**,**L**,**O**,**P**). Scale bars: **A**–**H**= 20 µm; **I**–**P**= 10 µm.

**Figure 5 ijms-24-00901-f005:**
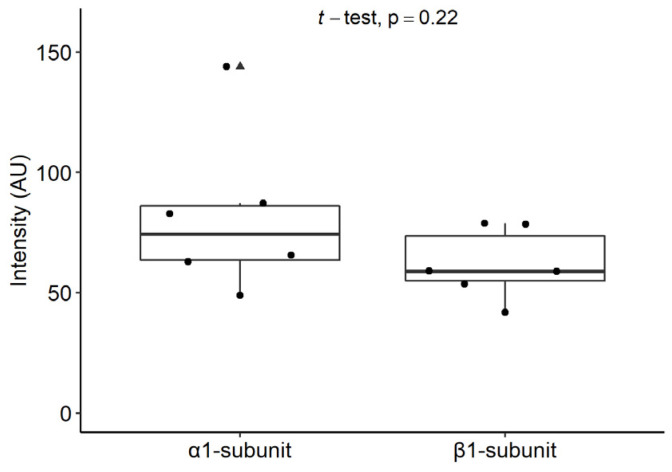
Staining intensities of the α_1_ and β_1_ subunits of sGC in the mast cells of inflamed human dental pulp. Note that there was no significant difference between the staining intensities of the α_1_ and β_1_ subunits of sGC in mast cells (*p* = 0.22). Box and whisker plot shows maximum (upper vertical line), minimum (lower vertical line), median (horizontal line), and interquartile range (box). Dots are single measurements and triangles are outliers. A *p*-value < 0.05 was considered as significant.

## Data Availability

Data sharing is not applicable to this article.

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
