# Peer review of "Dental Pulp Inflammation Initiates the Occurrence of Mast Cells Expressing the α1 and β1 Subunits of Soluble Guanylyl Cyclase"

_ijms, 2023, doi:10.3390/ijms24020901_

Round 1
Reviewer 1 Report
Authors attempted to describe the occurrence of mast cells expressing α1- and β1-subunits due to dental pulp inflammation. There are few comments based on current manuscript.
1) Comment 1: Please review all of misspelling especially upper script/lower script letters.
2) Comment 2: Point no 2.2 missing from the manuscript. Kindly correct it.
3) Comment 3: Please add some references in the Materials and method section.
4) Comment 4: If possible, please put the materials and method section after the introduction section.
Author Response
Response to reviewer 1:
We thank the reviewer for her/his constructive and helpful comments.
1) Comment 1: Please review all of misspelling especially upper script/lower script letters.
1) We have checked and corrected the spelling errors as suggested by the reviewer.
2) Comment 2: Point no 2.2 missing from the manuscript. Kindly correct it.
2) Thank you for the comment on the error. These errors have been corrected in the manuscript.
3) Comment 3: Please add some references in the Materials and method section.
3) As suggested by the reviewer, we have mentioned the relevant references (references 49, 50, 74 and 77) in the material and methods section of the manuscript.
4) Comment 4: If possible, please put the materials and method section after the introduction section.
4) We have rearranged the order as suggested by the reviewer according to the IJMS manuscript guidelines.

Reviewer 2 Report
The authors introduced that mast cells have a role in dental pulp immunity through sGC. The concept in the manuscript is interesting, however a large amount of fundamental information, including controls to prove the author's hypothesis, is missing.
In figure 2, please add arrows to emphasize the MCT.
A stained healthy sample to compare to an inflamed sample for MCT amount is missing in figures 2, 3, 4.
The quantifications are missing in figures 2, 3, 4.
In figures 3 and 4, higher magnification images are needed to see the location where the proteins are expressed.
The staining data showed that other immune cell types rather than mast cells are a major population in dental pulp. How do you explain that the mast cells can play a critical role in dental pulp immunity?
The authors mentioned in the introduction about ‘Tissue-resident mast cells', however the cells were not found in healthy condition. How can you call the mast cells “tissue-resident”?
There is no information about the NO measurement or readout of NO-GC signaling, for example cGMP production, cell proliferation etc, to prove the authors hypothesis.
Author Response
Please write down "Please see the attachment."

Round 2
Reviewer 2 Report
The authors did great improvement for the dataset, however there is a problem in the manuscript organization.
Please re-write the results and materials and methods section, and make sure results go to the result section and methods go to the materials and methods section.
Author Response
Response to reviewer 2:
We thank the reviewer for her/his constructive and helpful comments.
1. The authors did great improvement for the dataset, however there is a problem in the manuscript organization. Please re-write the results and materials and methods section, and make sure results go to the result section and methods go to the materials and methods section.
1. (a) In the Results section of the manuscript, the information described in lines 98-102 is a repetition of information already described in the Material and Methods section. Therefore, the sentences in lines 98-102 have been removed from the manuscript.
(b) To give the reader a general overview of the results, three new sentences have been added to the results.
(c) The legend of Figure 5 has been improved.
